# Establishment of DNA methylation during primate germ cell development

Kazuaki Kojima[1,2], Yi Li [3], Shin-ichi Tomizawa [4], Miho Kohara[5], Yuta Kuze[6], Takuya Sato[6], Toshinobu Ebata[6], Musashi Kubiura-Ichimaru[7], Mei Cao[8], Tatsuya Hattori [9], So Maezawa [9], Sofia B. Winge [10], Kristian Almstrup [10,11], Ryo Nakaki[12], Shinya Sato[13], Yohei Miyagi [13], Yataro Daigo[14,15], Takako Yoshioka[16], Yuichi Hasegawa[17], Yoshiaki Kinoshita[18], Taiju Hyuga[19], Kimihiko Moriya[19], Hideyuki Kobayashi[20], Laurence Baskin[8], Tadashi Sankai [5] & Toshiaki Watanabe [1,2] ✉

DNA methylation is almost completely erased throughout the genome in primordial germ cells, and then reestablished during mammalian germ cell development. In this study, we demonstrate that in three primate species—marmosets, macaques, and humans—de novo methylation occurs postnatally in prospermatogonia in males and growing oocytes in females. In monkey prospermatogonia, de novo methylation is a prolonged process spanning 6 months to 1 year, primarily occurring within the first year after birth. In human testes, this process may occur more slowly over an extended period. Single-cell bisulfite sequencing analyses in spermatogonia of three species revealed that all genomic regions acquire DNA methylation gradually. However, DNA methylation levels increase faster in genic regions compared to intergenic regions. Unlike in mice, mitotic divisions occur during the establishment of methylation in prospermatogonia. The established methylation is likely maintained because maintenance methyltransferase DNMT1 is specifically expressed during the mitotic stage. Our findings show notable differences in the de novo DNA methylation processes in male germ cells between mice and primates.

During germ cell development, DNA methylation is erased and reestablished. The reestablished DNA methylation is crucial for retrotransposon suppression and genomic imprinting[1,2]. The process has been extensively studied in mice. In male mouse germ cells, DNA methylation is established within a brief period from embryonic day 15 (E15.5) to 1 day after birth in the prospermatogonium stage. In female germ cells, it occurs during oocyte growth within developing follicles.

Molecular mechanisms and chromatin dynamics of DNA methylation establishment are well-characterized in mouse germ cells. de novo DNA methyltransferases DNMT3A, 3B, and 3 C catalyze methylation at CpG cytosines, with distinct genomic region preferences[3–5]. DNMT3A methylates most of the genome, while DNMT3B and DNMT3C target active retrotransposon promoters and centromere repeats. Primates lack DNMT3C, which evolved from DNMT3B in muroid rodents. Maintenance DNA methyltransferase DNMT1 adds methyl groups to newly synthesized DNA strands to maintain the already established methylation pattern. During de novo methylation in germ cells, mouse germ cells do not divide, suggesting DNMT1 does not contribute to the establishment[6,7].

PIWI-interacting RNAs (piRNAs) are 25–30 nt small RNAs. piRNAs are incorporated into PIWI proteins and mediate epigenetic silencing during DNA methylation establishment in male germ cells. The primary targets of piRNAs are active retrotransposons that are normally repressed by methylation[8–10]. DNA methylation of retrotransposon

---

sequences is similarly erased and reestablished during germ cell development. This reestablishment of retrotransposon DNA methylation occurs in a manner dependent on PIWI-piRNAs. PIWIL4 (also known as MIWI2 in mice) drives these epigenetic functions, while other PIWI proteins (PIWIL1 and PIWIL2) regulate the posttranscriptional processes. PIWIL4 uses piRNAs as guides to target nascent RNAs[11] and interact with SPOCD1, linking them to epigenetic functions through DNMT3A/3B/3 C and chromatin remodeling complexes like NURD and BAF[12]. This pathway mediates DNA methylation at the later stage during the establishment process[13].

Additionally, differentially accessible domains (DADs) in mouse prospermatogonia acquire DNA methylation later than other regions. In DADs, increased chromatin accessibility coincides with the initiation of DNA methylation establishment[14], suggesting accessibility is necessary for methylation. However, the extent to which these mechanisms are conserved in primates remains unclear.

In humans, DNA methylation in the female germline is established during oocyte growth[15]. However, the timing of this process in the human male germline remains unclear. In marmoset monkeys, DNA methylation establishment in male germ cells occurs postnatally during the prepubertal stage[16]. It begins by 4 months of age and continues until puberty at approximately 8 months[16]. As puberty timing differs between marmosets and humans, the human establishment stage cannot be directly inferred from marmosets. As such, insights from other primates with extended prepubertal stages could help extrapolate these findings to humans.

Here, we investigated the timing of de novo methylation in the germlines of three primate species (marmosets, cynomolgus monkeys, and humans). Additionally, we analyzed the complete de novo methylation processes in testicular germ cells using single-cell bisulfite sequencing (scBS-seq) in the three species. Our study highlights both similarities and differences in de novo DNA methylation between mice and primates.

## Results

### Establishment of DNA methylation in primate germ cells

To pinpoint the timing of DNA methylation establishment in primate germ cells, immunohistological analysis was performed using an antibody specific to 5-methylcytosine (5mC) in the ovaries of two primate species: cynomolgus monkeys and marmosets. In both species, DNA methylation signals were absent in oocytes of primordial and primary follicles (Fig. 1A, B). Most oocytes in secondary follicles displayed signals in both species (Fig. 1A–D). A similar pattern was observed in ovaries from both young and aged monkeys (Fig. 1C, D). These findings indicate that DNA methylation is established in oocytes of secondary follicles. The stage of DNA methylation in oocytes hence appears to be largely conserved between mice and monkeys[17].

In mice, DNA methylation in male germ cells occurs between E15.5 and 1 day after birth. Germ cells in this period, termed prospermatogonia, later differentiate into spermatogonial stem cells within several days after birth. In primates, primordial germ cells differentiate into prospermatogonia during midgestation and remain in this stage until puberty. As puberty in primates occurs much later after birth, primates have an extended postnatal prospermatogonium phase. To approximate the timing of de novo methylation in primate testes, immunohistochemical analyses were conducted. In accordance with Langenstroth-Rower et al.[16], 5mC signals in marmoset testicular germ cells were absent or faint at 21 days of age, varied at 3 months, and strong at 9 months (Fig. 1E). Similarly, in cynomolgus monkeys, testicular germ cells showed no or weak 5mC signals at 29 days but displayed strong signals at 6 months (Fig. 1F). These observations indicate that de novo DNA methylation in these primates occurs postnatally, predominantly within several months after birth.

### Single-cell analyses of de novo methylation in marmosets

To investigate de novo methylation dynamics with greater precision during marmoset postnatal development, scBS-seq analysis was performed alongside simultaneous scRNA-seq analyses (see later) from the same cells. A total of 103 FACS-purified marmoset germ cells, spanning 10 developmental stages (1 day to 12 months of age), were sequenced (Supplementary Data 1, 2, and 3). Conversion rates were estimated to be 99.2%–99.4% using externally added unmethylated lambda DNA (Supplementary Data 3). Among the 103 cells analyzed, two exhibited low mapping rates (1% and 0.5%), and one was identified as a somatic cell based on DNA methylation patterns at imprinted loci (Supplementary Data 3, 4). After excluding these three, a total of 100 cells (4–16 per stage) were retained for analysis (Supplementary Data 2). The average genome coverage rate was 24.2%, ranging from 10.4% to 34.4% (see Supplementary Data 3 for detailed mapping statistics). The average DNA methylation level in 1-day-old germ cells (Fig. 2, and Supplementary Data 3) was at its minimum (4.9%), closely resembling the level reported for iPS cell-derived early prospermatogonia (4.3%) in a previous study[18]. This observation suggests that de novo methylation primarily occurs postnatally. A modest rise in DNA methylation was noted from 1 day (4.9%) to 1.1 months (10%) (Fig. 2). This gradual increase continued from 1.1 to 2.5 months (20%), although one 2.5-month prospermatogonia exhibited a markedly higher methylation level (61%). Following 2.5 months, the sharpest increase was observed up to 6.4 months (57%), during which varying methylation levels were detected even among prospermatogonia from the same testis (Fig. 2). Immunostaining analyses also revealed varied signal intensities among germ cells in 3-month-old testes (Fig. 1E). Beyond this period, DNA methylation levels continued to rise, with an additional 11.1% increase noted from pubertal 8.5-month prospermatogonia/spermatogonial stem cells (60%) to 12.5-month spermatogonial stem cells (71%) with potential for spermatogenesis. This level approached that of sperm (76%), as determined by bisulfite sequencing of bulk DNA.

### de novo DNA methylation patterns in the marmoset male germ cells

Substantial variation in DNA methylation levels was observed among germ cells within the same testes (Fig. 2), reflecting unsynchronized developmental progression. To examine methylation dynamics in detail, cells were grouped into seven categories (G1–G7: G1, 0%–10%, 11 cells; G2, 10%–20%, 16 cells; G3, 20%–30%, 8 cells; G4, 30%–40%, 10 cells; G5, 40%–50%, 21 cells; G6, 50%–60%, 19 cells; G7, 60%–80%, 15 cells) according to total DNA methylation levels (Fig. 2). Most G1 and G2 cells originated from samples aged 1 day to 2.5 months. G3–G5 were predominantly from 3.4–5 months, while G6 and G7 were mainly derived from 6.4–8.5 and 12.5 months, respectively.

Gradual establishment of global DNA methylation raises two possibilities regarding the pace of local methylation acquisition: (1) DNA methylation is acquired quickly (e.g., within one week, as observed in mouse prospermatogonia) at the regional level but the establishment occurs at different developmental stages depending on genomic region (fast pattern, Fig. 3A top), or (2) methylation occurs gradually at the regional level as well (gradual pattern, Fig. 3A bottom). To discern these patterns, the genome was divided into 500-kbp tiles, and the average methylation level for each group was calculated within each tile, with results displayed along each chromosome (Fig. 3B, Supplementary Fig. 1). DNA methylation levels were gradually increased across genomic regions (Fig. 3B), supporting the gradual pattern. However, close inspection revealed that the final (G7) levels varied significantly among regions. Furthermore, slight differences in establishment kinetics between regions were revealed by comparing methylation acquisition between early (G3–G1, red line in Fig. 3B top) and late (G7–G5, blue line in Fig. 3B top) stages (Fig. 3B). Thus, at the

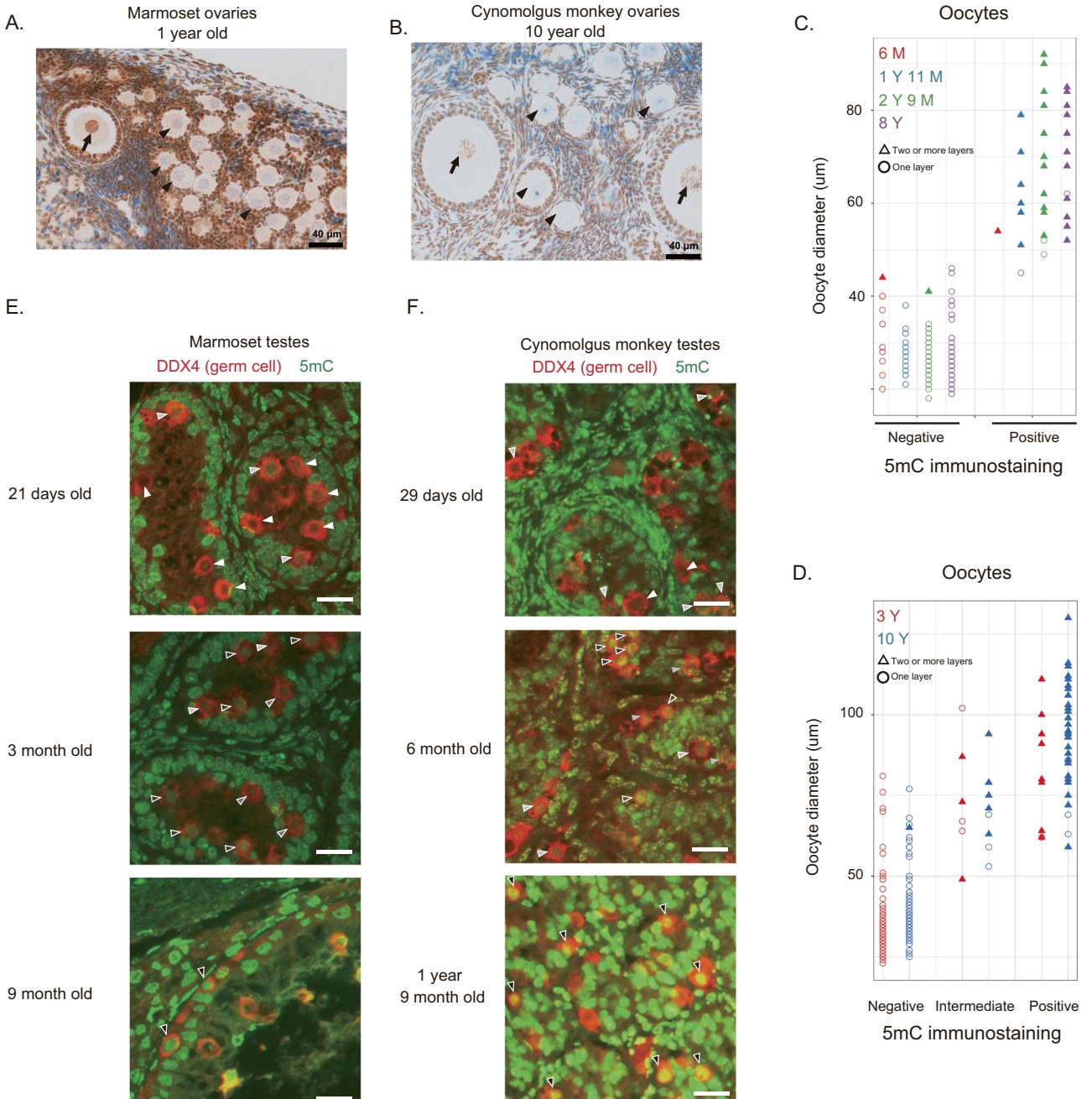

**Fig. 1 | Stages of de novo DNA methylation in the monkey germline.**
**A** Immunohistochemical analysis of marmoset ovaries shows that oocytes in secondary follicles exhibit a 5mC signal (arrow), whereas oocytes in primordial and primary follicles lack signals (arrowheads). The results were reproduced in three other animals of different ages (see **C**). **B** Immunohistochemical analysis of cynomolgous monkey ovaries using the 5mC antibody highlights similar patterns. The result was reproduced in another animal (see **D**). **C**, **D** The relationship between oocyte diameter and methylation levels is illustrated, with open circles (filled triangles) representing oocytes surrounded by one layer (multiple layers) of follicle cells. Source data are provided as a Source Data file. **E**, **F** Immunofluorescence analysis of marmoset (**E**) and cynomolgus monkey (**F**) testes to determine methylation levels in germ cells (arrowheads). The intensity of 5mC signals in germ cells is represented by the color of the arrowheads: white (weak), gray (intermediate), and black (strong). Each result was confirmed at least once using other sample(s) with similar age(s). Since the intensities of DNA methylation signals are highly sensitive to the quality of the samples (e.g., differences in fixative and storage conditions), the results of low-quality samples were not considered. Scale bars are 20 μm.

local level, de novo methylation proceeds gradually, but establishment kinetics slightly differs among genomic regions.

Some genomic regions exhibited preferential DNA methylation acquisition during the early stage, reaching a plateau more rapidly (Fig. 3B, red arrows), while others demonstrated a slower and continuous increase until the later stages (Fig. 3B, black arrows). To elucidate the regional differences in DNA methylation establishment patterns, genomic tiles were categorized into three groups (fast, intermediate, and slow) based on the speed of methylation acquisition. This categorization was achieved by comparing methylation differences between the early two stages (G3–G1) and the final two stages (G7–G5) (Fig. 3C; see "methods"). Slow regions, characterized by lower methylation in (G3–G1) than in (G7–G5), were predominantly associated with intergenic regions and retrotransposons enriched in

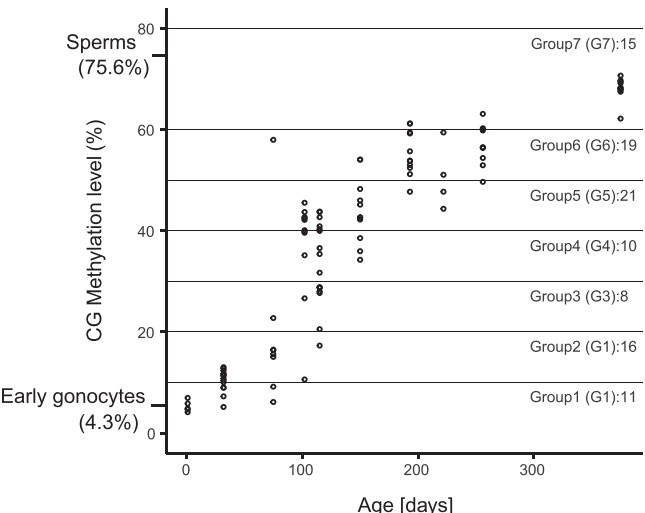

**Fig. 2 | De novo DNA methylation in the marmoset testicular germ cells.** The DNA methylation levels during postnatal male germ cell development are depicted, with each point representing the average methylation level of a single cell as determined by scBS-seq. The methylation levels of early gonocytes derived from iPS cells[18] and sperm from adult testes are also shown. Cells were grouped according to DNA methylation levels, with the number of cells in each group indicated to the right of the group labels.

intergenic regions (LINE and LTR elements) (Fig. 3B, D, and Supplementary Figs. 1, 2A, B). Conversely, fast regions were predominantly associated with genic regions, including promoters, exons, and introns, along with annotations enriched in these regions, such as ATAC peaks and CpG islands (Supplementary Figs. 2A, B). These findings reveal that, at the macroscopic scale, genic regions seem to more easily acquire DNA methylation than intergenic regions (Fig. 3E).

The final DNA methylation levels varied significantly between genomic regions. To elucidate the basis for these differences, average methylation levels were assessed across structural annotations (exons, introns, intergenic regions, flanking regions, and promoters) and functional annotations (CpG islands and ATAC peaks) (Fig. 3F, and Supplementary Figs. S2–4). Exons (77%) and introns (79%) showed higher methylation levels than intergenic (72%) and gene flanking (65%) regions in G7 (Fig. 3F). However, although active regulatory elements, such as promoters (excluding low CpG promoters), CpG islands, and ATAC peaks from testicular germ cells at 1 and 3 months of age, were mainly located in genic regions (Figures S2A, B), they were essentially devoid of methylation (Fig. 3F, G). Notably, when active regions (promoters, CpG islands, and ATAC peaks) and their surrounding sequences (± 2 kb from the center) were excluded (Fig. 3G, bottom), the genome-wide average methylation level increased by approximately 10%. The enrichment of these active regions within genic regions (Figures S2A, B) suggests that genic regions have a stronger innate tendency to acquire DNA methylation than indicated by this analysis.

### Mechanisms of de novo DNA methylation in marmoset male germ cells

**Association of active transcription with efficient DNA methylation.** Previous studies in mouse oocytes have shown that DNA methylation establishment at genic regions depends on transcriptional activity[19,20]. To test whether transcription similarly influences the rapid kinetics of methylation at genic versus intergenic regions in primates, we performed correlative analyses of scBS-seq, scRNA-seq, and ATAC-seq data from marmoset prospermatogonia. Genes were divided into three groups—high, mid, and low—based on expression levels, and DNA

methylation at gene bodies was compared (Fig. 3H). Across all groups (G1–G7), methylation levels were positively correlated with mRNA expression levels. We further classified genes into fast, intermediate, and slow categories according to the speed of methylation acquisition in gene bodies, then examined promoter ATAC peaks (Fig. 3I). This revealed clear differences in promoter accessibility: fast genes displayed the highest accessibility, followed by intermediate and slow genes. Together, these findings indicate that efficient methylation acquisition at genic regions in primate prospermatogonia depends on transcription of genes with accessible promoters.

**Expression of the components involved in de novo DNA methylation.** DNMT3A and DNMT3B catalyze the transfer of methyl groups onto unmethylated cytosines in CpGs with the assistance of catalytically inactive DNMT3L[4,5,21]. The expression of these de novo DNA methyltransferases was investigated through simultaneous scRNA-seq data analysis. Average expression levels were assessed across groups (G1–G7) categorized by DNA methylation levels (Fig. 4A). Non-smooth expression patterns across development were likely due to the limited number of cells per group. During de novo DNA methylation in marmoset male germ cells, DNMT3A expression was detected across all groups, whereas DNMT3B and DNMT3L expression was primarily restricted to the later groups (Fig. 4A, B). We next performed immunofluorescence analyses of the three DNMT3 proteins. In neonatal testes, DNMT3A was detected in the nuclei of a small subset of germ cells, while many showed weak cytoplasmic staining (Fig. 4C). Similar weak cytoplasmic staining was also observed for DNMT3B, whereas DNMT3L expression was absent. By 3 months of age, all three DNMT3 proteins exhibited clear nuclear staining in a fraction of germ cells, though signal intensities varied and some cells lacked expression entirely. At 9 months, DNMT3A and DNMT3L were expressed in ~90% and ~20% of spermatogonial stem cell-like cells, respectively. DNMT3B showed weak cytoplasmic expression in a small subset of these cells. Additionally, DNMT3A and DNMT3B expression was observed in Sertoli cells of 9-month-old testes. In mouse prospermatogonia, the PIWI-piRNA pathway regulates de novo methylation of retrotransposons[8,9]. Similarly, expression of key PIWI-piRNA-mediated DNA methylation components, PIWIL4, SPOCD1, and TEX15, was observed during de novo methylation stage in marmosets (Fig. 4A, B).

**Cell division during de novo DNA methylation.** Unlike the quiescent G0 state of mouse prospermatogonia during de novo DNA methylation, our simultaneous scRNA-seq analyses revealed that marmoset prospermatogonia undergo cell division (Fig. 4B). Of the 88 cells that were in the de novo methylation process (methylation level: 10% − 70%), 11 cells (DNA methylation levels: 12%, 12%, 40%, 40%, 43%, 46%, 56%, 60%, 62%, 68%, 70%) were likely in the mitotic phases (S and G2/M) (Fig. 4B, Supplementary Data 3). This prompts us to question whether the amount of the established DNA methylation is decreased due to the division. Interestingly, DNMT1 expression is observed exclusively in the dividing cells, whereas DNMT3A/B expression is mainly observed in quiescent cells (Fig. 4B, D). This differential DNMT1 expression pattern suggests that the established methylation is likely maintained after cell division, during the establishment stage.

**Non-CG methylation.** A previous study in mice reported markedly high levels of non-CG methylation (~8% of all non-CG sites), peaking in neonatal prospermatogonia[22]. To assess whether this is conserved across species, we analyzed CH methylation (H = A, C, or T) in marmoset prospermatogonia. Quantification showed that the fraction of methylated cytosines among all CH dinucleotides ranged from 0.7% to 1.1% during the G1–G7 transition, in contrast to the sharp increase in CG methylation (Fig. 4E). In G7, ~4.5% of all cytosines were methylated (vs. 10.8% in mouse neonatal testis germ cells[22]), with 3.5% (3.3% in mice) at

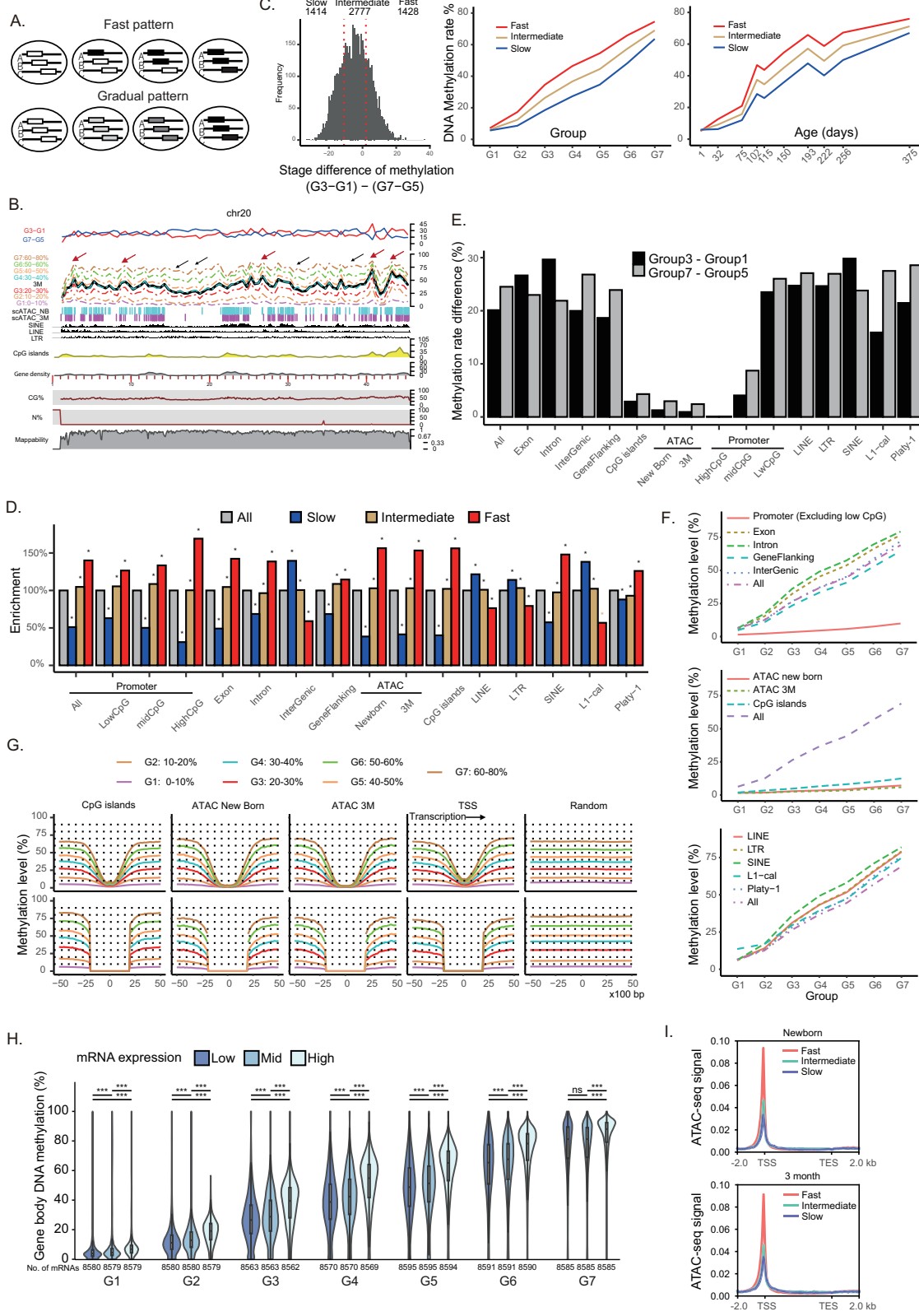

CG sites and 1.0% (7.5% in mice) at CH sites. CH methylation occurred most frequently at CA dinucleotides, particularly at CAC and CAG trinucleotides, which showed an increasing trend across G1–G7 (Fig. 4F, left). Since CAC and CAG are preferred targets of DNMT3A and DNMT3B, respectively[23–25], these results indicate that CH methylation depends on the expression of these enzymes. For CG methylation, CGC and CGG are reported targets of DNMT3A and DNMT3B[25]. These

trinucleotides were enriched among methylated CpG sites but showed a decreasing trend from G1 to G7 (Fig. 4F, right), likely reflecting preferential methylation at early stages and reduced availability later. Collectively, these findings demonstrate that CH methylation levels are lower in marmosets than in mice, likely due to the active proliferation of marmoset prospermatogonia, as non-CG methylation is not maintained during DNA replication.

**Fig. 3 | De novo DNA methylation patterns in different genomic contexts.**
**A** Extreme models for the acquisition patterns of methylation during de novo methylation. **B** Genomic annotations and patterns of methylation establishment along chromosome 20 are presented. The region's mappability, determined using Bismap with an input length of 50 nt, is shown. Cells are classified into seven groups (G1–G7) based on their average methylation rates, with group-specific average methylation levels plotted in different colors. Regions acquiring methylation at relatively fast speed (red arrows) and at slow speed (black arrows) are highlighted. Differences in methylation levels between G3 and G1 (G7 and G5) are shown as red (blue) lines, respectively. scATAC-seq data for newborn (NB) and 3-month-old (3 M) testes germ cells are also included. **C** Genomic tiles (500 kbps) are categorized into three classes−fast, intermediate, and slow−based on the patterns of DNA methylation establishment. Average methylation patterns of the three classes are shown by group and age. **D** Enrichment of specific genomic annotations in the three classes (slow, intermediate, fast) is demonstrated. Asterisks show statistical

significance ($P < 0.05$) by the Chi-squared test. **E** Differences in methylation levels during the early (G3–G1) and late (G7–G5) phases are quantified. **F** Average methylation patterns across genomic contexts are displayed. **G** Local methylation patterns are examined around CpG islands, ATAC peaks, and TSSs. Below, methylation levels are reanalyzed after removing the indicated annotation sequences and surrounding sequences ($\pm 2$ kb from the center of these elements). **H** Genes were equally divided into three subsets based on expression levels. P-values (two-sided Wilcoxon test) were adjusted for multiple comparisons using the Benjamini-Hochberg method (***$p < 0.05$; ns, not significant). Box plots show the median (center line), 25th and 75th percentiles (box limits), and whiskers extending to the most extreme values within $1.5 \times$ interquartile range (IQR). Violin plots are overlaid on the boxplot. **I** Genes were classified into three categories based on the speed of methylation acquisition in gene bodies. Source data are provided as a Source Data file for (**C**–**I**).

**Accessibility.** In mouse testes, certain genomic regions, termed DADs, resist methylation during the early phase of de novo methylation[14]. DADs acquire DNA methylation only after increased accessibility during the late de novo methylation stage[14]. To explore the presence of DAD-like regions in marmosets, scATAC-seq analyses were conducted on newborn and 3-month-old marmoset testes. Newborn and 3-month-old germ cells showed similar patterns of chromatin accessibility peaks (Fig. 3B, and Supplementary Figs. 1). Despite no clear differences in accessibility, the entire genomic regions of 3-month-old germ cells displayed increased DNA methylation to the level almost proportional to the final levels (Fig. 3B, and Supplementary Fig. 1). Consequently, DAD-like regions were not evident during the establishment phase in marmosets.

### Conservation of the timing of de novo DNA methylation in cynomolgus monkeys

scBS-seq analysis was performed on postnatal cynomolgus monkeys to accurately assess DNA methylation levels during male germ cell development. Thirty germ cells spanning seven developmental stages were analyzed (Fig. 5A; and Supplementary Data 1, 2, and 5). In 9-day prospermatogonia, the average DNA methylation level was very low (3.6%) (Fig. 5A, and Supplementary Data 5). Levels increased in 60-day prospermatogonia (28%) and further increased in 6-month prospermatogonia (58%), with substantial variation among cells at these stages. In 1.1-year prospermatogonia, the level nearly plateaued (70%), with only slight increases observed thereafter (72% in 1.6-year prospermatogonia, 70% in 1.7-year prospermatogonia). Adult spermatogonial stem cells (10 years) exhibited a methylation level of 75%. Thus, although puberty onset varies between marmosets (9 months) and cynomolgus monkeys (4 years), de novo methylation in male germ cells occurs at similar ages.

To examine the pattern of methylation establishment, 30 cells were grouped into five categories (G1–G5) according to total DNA methylation levels (Fig. 5B). Average methylation levels in each group were then plotted across 500-kbp genomic tiles. As in marmosets, DNA methylation appeared to accumulate gradually and in parallel across the genome (Fig. 5B). Analysis of non-CG methylation revealed that only 1.0%–1.8% of all CH dinucleotides were methylated (Fig. 5C), far lower than the ~8% observed in mouse neonatal prospermatogonia[22]. Preferential targeting of CAG/CAC and CGG/CGC trinucleotides was evident for CH and CG methylation, respectively (Fig. 5D). These findings indicate a conserved mode of non-CG methylation among primates but underscore a striking divergence from rodents.

To investigate genomic context-dependent methylation kinetics in cynomolgus monkeys, genomic tiles were categorized into three groups: fast, intermediate, and slow (Fig. 5B, E, and Supplementary Fig. 5). As observed in marmosets, annotations associated with genic and intergenic regions were enriched in fast and slow groups, respectively (Fig. 5F, and Supplementary Figs. 2B, 5). Additionally, for

the final methylation level (G5), genic regions (exon 80% and intron 82%) acquired higher methylation levels than intergenic (77%) and gene flanking regions (68%) (Fig. 5G, and Supplementary Fig. 5). Hence, the timing and pattern appeared consistent between cynomolgus monkeys and marmosets.

### Establishment of DNA methylation in human germ cells

By extrapolating findings from the two monkeys, establishment stages in humans were predicted to occur in secondary follicle oocytes and testicular germ cells between 0 and 1 years of age. Adult ovaries excised from uterine cancer patients (Supplementary Data 6) were analyzed via immunohistochemistry with 5mC antibodies. Although we cannot exclude the possibility that cancer tissues influence the developmental establishment of oocyte methylation, normal tissues resected from cancer patients provide valuable insights into human in vivo biology. Consistent with the findings in monkeys: no 5mC signals were detected in primary follicle oocytes (Fig. 6A), and weak signals appeared in secondary follicle oocytes, with stronger signals in antral follicle oocytes. Additionally, these results align with bisulfite analyses by Yan et al., demonstrating variable levels of DNA methylation in human growing oocytes[15].

To examine the timing of de novo methylation during human male germ cell development, immunofluorescence analyses using a 5mC antibody were performed on aborted normal gonad (E21-week), testicular biopsies from patients with severe combined immunodeficiency (for fertility preservation, 3 months), medulloblastoma (for fertility preservation, 7 months), and unilateral undescended testis (biopsy of normal testis, 8 years) (Fig. 6B, and Supplementary Data 6). Because these diseases were observed outside the examined testis, they most likely had little effect on the results. In addition, we examined surgically removed testes from testicular teratoma patients (5 months and 1-year 5-months) (Fig. 6C, and Supplementary Data 6). Because the testicular tissues analyzed were directly in contact with the teratomas, their developmental progression may have been partially influenced by the tumor environment. No or very weak methylation signal was detected in germ cells from the aborted fetus (E21). Weak signals were observed in 3-month germ cells of biopsy samples. However, 5-month germ cells (surgical sample from a teratoma patient) showed no or very weak signals. Clear, although not strong, signals were present in some germ cells in 7-month testes (biopsy). Stronger 5mC signals were observed in most germ cells at 1 year and 5 months (surgical samples from teratoma patients), with levels almost comparable to those in 8-year testis germ cells (biopsy). However, some germ cells in 8-year sample showed nearly completely negative staining (Fig. 6B right, yellow arrows). This is possibly due to unfavorable cellular conditions for 5mC immunostaining (e.g., degradation of DNA, demethylation, difficulty in accessing epitopes), as many Sertoli cells also showed negative staining. Similar negative staining was observed in testes after 9 months of storage, although this was not

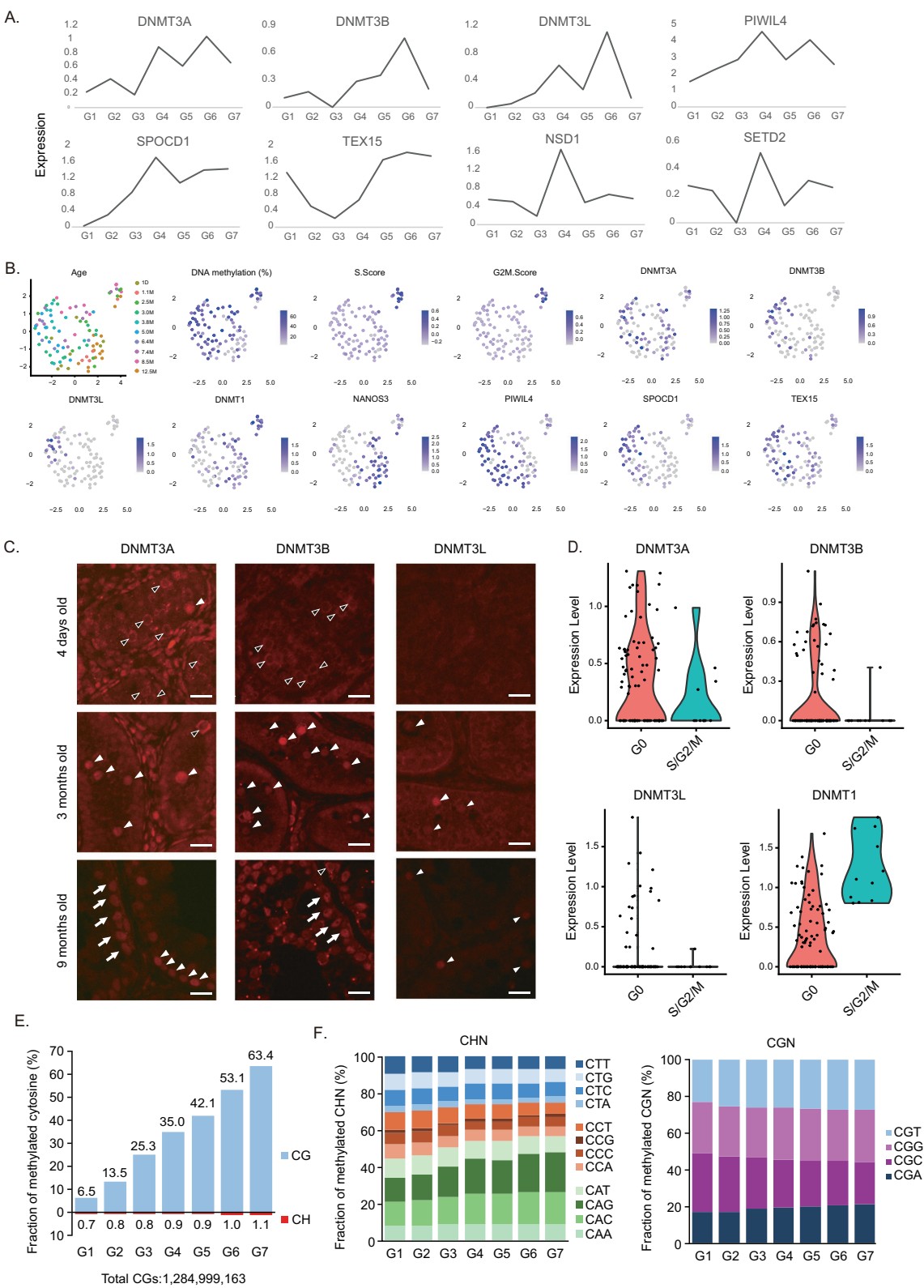

observed when the same sample (a 1-year-5-month testis from a teratoma patient) was examined immediately after collection (Supplementary Fig. 6). Considering considerable variations in sample state (e.g., fixative, storage time, storage conditions, and disease) (Supplementary Data 6), accurately comparing methylation levels across samples is difficult. However, the data provide some preliminary information about the timing. The timing of de novo methylation in human germ cells hence seems slightly later than that in marmosets and cynomolgus monkeys.

To quantitatively assess methylation levels, single-cell bisulfite sequencing was carried out on human germ cells. Prospermatogonia from three teratoma patients (1-year-0-month, 1-year-5-months (also examined in Figs. 6C), and 1-year-8-months) and spermatogonial stem cells from a 29-year-old obstructive azoospermia patient with normal

**Fig. 4 | Gene expression and mitotic activity during DNA methylation establishment. A** Expression profiles of de novo DNA methylation components during the establishment phase, analyzed using multiome (scRNA-seq and scBS-seq) data. Cells are grouped into seven classes based on average methylation levels, as described in Fig. 3B. LogNormalize function of Seurat was used for normalization. The average expression levels of cells in the same group are shown. **B** Mitotic activity during de novo DNA methylation. scRNA-seq data were analyzed by UMAP. Ages of animals where cells are derived, DNA methylation levels (determined via simultaneous scBS-seq analyses), mitotic indexes, and expression levels of de novo methylation genes are visualized. **C** Immunofluorescence analyses of marmoset testes at three developmental stages (4 days, 3 months, and 9 months) showing DNMT3 protein expression. White and black arrowheads indicate germ cells with

nuclear and cytoplasmic staining, respectively. White arrows in 9-month-old testes mark nuclear staining in Sertoli cells. The staining patterns were reproducible using the same samples. Scale bars, 20 μm. **D** Expression of DNMTs in mitotically inactive and active cells. The expression levels of the 100 cells are shown. Cells were classified into G0 (88 cells) and S/G2/M (12 cells) groups according to the results in (**B**). **E** Fraction of methylated cytosines at CG (upper) and CH (lower) dinucleotides in prospermatogonia across G1–G7 groups. Numbers below each plot indicate the total CGs and CHs (including both methylated and unmethylated) analyzed from scBS-seq data. **F** Classification of methylated CHN (left) and CGN (right) trinucleotides into specific sequence motifs. Source data are provided as a Source Data file for (**A**, **B**, **E**, and **F**).

spermatogenesis were analyzed (Supplementary Data 6). The three 1-year-old prospermatogonia exhibited average methylation levels of 41.4% (23.1–54.4%, 1-year-0-month), 53.1% (32.0–62.4%, 1-year-5-months), and 32.6% (10.2–54.7%, 1-year-8-months) (Fig. 7A, Supplementary Data 7). No correlation between methylation levels and ages in these 1-year-old samples is likely due to individual variation and/or effects of teratoma. In contrast, adult spermatogonia showed an average level of 75.7% (73.3–76.8%). Although teratoma may have influenced the developmental timing of prospermatogonia within the same testes, prospermatogonia across all 1-year-old samples remained in an intermediate phase of DNA methylation establishment.

To investigate regional establishment patterns, single cells were sorted into seven groups (G1–G7) based on methylation levels. Genomic tile (500-kbp) analyses revealed that DNA methylation levels were gradually increased across genomic regions (Fig. 7B, Supplementary Fig. 7). The methylation levels of non-CpG sites in human prospermatogonia (1.5–2.1%, G1–G6 in Fig. 7C) were slightly higher than those in monkey prosermatogonia (marmoset: 0.7%–1.0%; cynomolgus: 1.0%–1.8%), yet remained lower than in mouse prospermatogonia (~8%)[22]. Moreover, human spermatogonial stem cells (G7 in Fig. 7C) displayed elevated methylation not only at CpG sites (75.7%) but also at non-CpG sites (5.4%) compared with marmoset G7 (CpG: 63.4% and non-CpG: 1.1%) and cynomolgus monkey G5 (CpG: 64.0% and non-CpG: 1.5%), which include spermatogonial stem cells. This pattern indicates robust de novo methylation activity in human spermatogonial stem cells. Genomic tiles were sorted into three classes (slow, intermediate, fast) according to the acquisition pattern of DNA methylation, and genomic-context-dependent methylation kinetics were evaluated. As in monkeys, genic (intergenic) regions tended to gain methylation faster (slower) in development (Fig. 7B, D, E, F). Among the analyzed sequences, introns exhibited the highest levels in prospermatogonia (G1–G6 in Fig. 7F), whereas LINE sequences were most highly methylated (85.3%) in spermatogonial stem cells (G7 in Fig. 7F).

## Discussion

This study sought to define the time window and establishment pattern of de novo methylation in primate germ cells. In oocytes of all three primate species studied, de novo methylation begins in secondary follicles. In testicular germ cells of the two monkeys, de novo methylation starts postnatally, continues for over 6 months, and is nearly complete within one year of birth. Although a future sequencing study using samples from non-teratoma patients is needed to confirm, this process appears to occur slightly later in human testicular germ cells. scBS-seq analyses revealed that methylation establishment initiates simultaneously across all genomic regions. However, genic regions are methylated faster and reach a plateau earlier than intergenic regions in all three species. Unlike in mice, primate prospermatogonia undergo cell divisions, with specific DNMT1 expression in mitotic cells, suggesting that methylation is maintained during division. This study highlights differences in de novo DNA methylation dynamics between mouse and primate prospermatogonia.

Although the entire genome undergoes de novo methylation simultaneously and gradually, genic regions, in particular highly expressed genes (Fig. 3H, I), acquire methylation slightly faster than intergenic regions. This difference may be attributed to the relative ease of physical interaction with DNA methyltransferase enzymes, as genic regions are more accessible than intergenic regions. Moreover, differential expression of DNMT3A and DNMT3B (Fig. 4A–C) may partly contribute to these variations, given that DNMT3B preferentially targets heterochromatin and repeat regions, unlike DNMT3A[4,5,21]. In mouse prospermatogonia, H3K36 methylation deposited by NSD1 and SETD2 directs DNA methylation establishment[26]. SETD2 deposits H3K36me3 on actively transcribed genes, whereas NSD1 deposits H3K36me2 across broad domains. Thus, actively transcribed regions, as redundant targets of both enzymes, acquire both H3K36me2 and H3K36me3 marks. H3K36me3 enrichment at these regions may play an additive role in promoting DNA methylation. In marmoset prospermatogonia, SETD2 (along with NSD1) is expressed (Fig. 4A), suggesting that SETD2- mediated H3K36me3 may contribute to the fast methylation of active regions.

Impairment of DNA methylation establishment in retrotransposon sequences has been linked to developmental arrest in mouse adult testes[8]. A recent study by Zoch et al.[27]. reported that loss-of-function variants in SPOCHD1 lead to male infertility and LINE1 upregulation in spermatogonial stem cells. In support of this, our findings in marmosets revealed that prospermatogonia at the stage of de novo methylation express PIWIL4, SPOCHD1, and TEX15, key components of the PIWI-piRNA-mediated DNA methylation in mice. Further studies are needed to confirm whether the PIWI-piRNA pathway also mediates retrotransposon DNA methylation in primates.

Cancer treatments involving chemotherapy and radiation often result in male infertility. While adult men can preserve fertility by banking sperm before treatment, prepubertal boys currently do not have options for fertility preservation. For these patients, experimental oncofertility programs can offer cryopreservation of testicular tissue (https://oncofertility.msu.edu, https://www.orchid-net.com). However, methods to induce spermatogenesis in cryopreserved prepubertal human testes remain undeveloped. In cynomolgus monkeys, healthy offspring have been produced using sperm derived from cryopreserved prepubertal testes through xenotransplantation to nude mice or autotransplantation[28,29]. In xenotransplantation studies, testes from 14- and 27-month-old monkeys were transplanted under the back skin of castrated nude mice, with functional sperm observed 10 months post-transplantation[28]. In autotransplantation studies, cryopreserved testes from 33-month-old monkeys were autotransplanted under the back skin of castrated monkeys, with functional sperm similarly observed within 8–12 months[29]. Based on our findings, it is likely that the germ cells in these transplanted testes were fully methylated, enabling spermatogenesis. Methylation levels in human germ cells serve as an indicator of developmental progression, which is a key component in furthering research in human spermatogenesis for oncofertility patients.

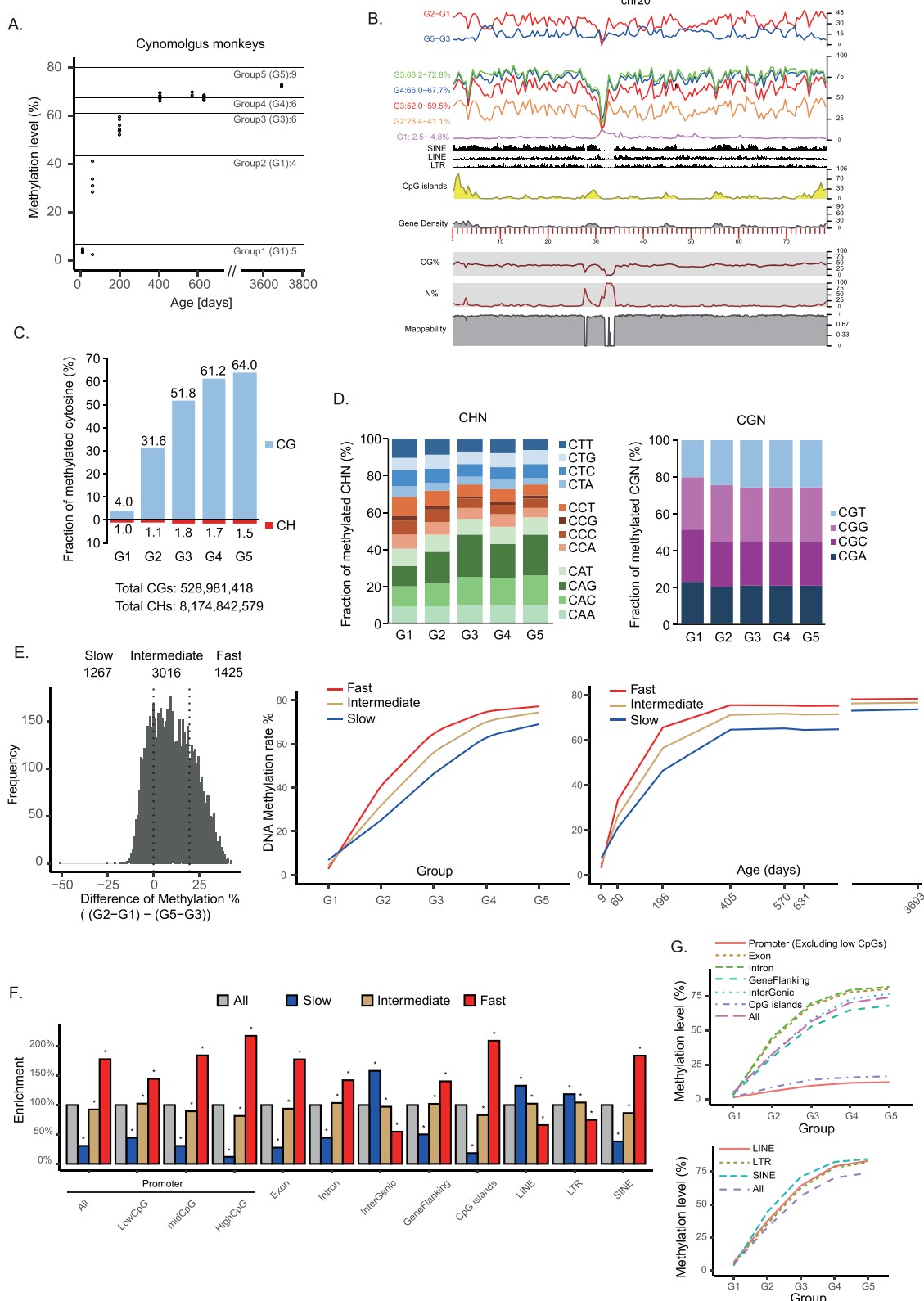

The next challenge for transplantation studies in monkeys is to test younger testes prior to de novo methylation. We hypothesize that unmethylated prospermatogonia require additional time to undergo de novo methylation before developing into spermatogonial stem cells. Understanding the mechanisms that regulate the initiation and pace of de novo methylation could accelerate sperm production from unmethylated testes.

De novo DNA methylation in primate male germ cells requires several months or more than one year, contrasting with male mouse germ cells, where it occurs within a week. Possible explanations for these species-specific differences include (1) variations in DNMT3 enzyme activity (amino acid sequences), (2) differing levels of DNMT3 expression, (3) variations in genome accessibility, and (4) differences in concentration of

**Fig. 5 | De novo DNA methylation in cynomolgus monkey male germ cells.**
**A** DNA methylation levels during postnatal male germ cell development are shown, with each point representing the average methylation level of a single cell determined by scBS-seq. **B** Genomic annotations and patterns of DNA methylation establishment along chromosome 20 are presented. **C** Fraction of methylated cytosines at CG (upper) and CH (lower) dinucleotides in prospermatogonia across G1–G5 groups. **D** Classification of methylated CHN (left) and CGN (right) trinucleotides into specific sequence motifs. **E** Different de novo DNA methylation kinetics are illustrated. Genomic tiles (500 kbps) are categorized into three classes (slow, intermediate, fast) based on establishment patterns ((G2–G1) − (G5−G3), left chart). Average methylation patterns of the three classes are displayed in relation to group (center) and age (right). **F** Enrichment of specific genomic annotations (e.g., CpG islands, TSSs) within the three classes is presented. Asterisks show statistical significance ($P < 0.05$) by the Chi-squared test. **G** Average patterns of increasing DNA methylation levels across different genomic contexts. Source data are provided as a Source Data file for (**C**–**G**).

S-adenosylmethionine (SAM), the methyl donor for DNA methylation.

An intriguing species difference revealed here is the apparent absence of DAD-like domains in primates[14]. In mouse prospermatogonia, DADs are found in heterochromatic regions characterized by low chromatin accessibility and resistance to methylation in early stages. In mid prospermatogonia, accessibility increases, and these regions subsequently undergo methylation. A possibility as to why DAD-like domains are not observed in primates is that mitosis in primate prospermatogonia substitutes for the activation mechanisms seen in mouse DADs—for instance, DNA replication and mitotic chromosome formation may facilitate chromatin reprogramming. Alternatively, DAD-like regions may exist in primates but were not detectable in this study due to the limited read depth of our ATAC data.

Another notable species difference is the timing of de novo DNA methylation (monkeys: postnatal; mice: pre- and postnatal). This divergence likely reflects differences in reproductive strategies. Mice accelerate germ cell development to enable reproduction shortly after birth, making germ cells at birth more advanced in mice than in primates. Despite this temporal discrepancy, the mechanisms initiating de novo methylation may be conserved across species. Future research should identify the cues triggering de novo methylation.

Understanding the mechanisms of DNA methylation establishment potentially helps in recapitulation of germ cell development from iPS cells and generation of sperm from cryopreserved infant testes.

## Methods

### Testis samples
Testis sampling was approved by the animal ethics committees of the National Center for Child Health and Development (NCCHD), CLEA Japan, Central Institute of Experimental Animals, and Tsukuba Primate Research Center (NIBIOHN). Details of the marmosets and cynomolgus monkeys used in sequencing analyses are provided in Supplementary Data 1. Samples were obtained via hemicastration under anesthesia or from monkeys euthanized for other purposes.

Details of human samples used in this study are summarized in Supplementary Data 6. The study's experiments were approved by the ethics committees of the NCCHD (2022-138 and 2024-044), Kanagawa Cancer Center (2022EKI97), and University of California San Francisco (#23-39078). This study was conducted in accordance with all relevant ethical regulations concerning research involving human samples.

### Histological examination
Testes were fixed in 4% PFA, 10% formalin, or Bouin's solution, embedded in paraffin, and sectioned. Antigen retrieval was performed using a citrate buffer pH 6.0 (5mC, MAGEA3/4, DDX4, and DNMT3B) or a Tris buffer pH 9.0 (DNMT3A and 3 L) at 100 °C for 15 min. Blocking was conducted with BOND Primary Antibody Diluent (DNMT3L and MAGEA3/4) or 5% skim milk/0.1% Tween 20/PBS (5mC, DDX4, DNMT3A, and 3B). Antibody reactions utilized BOND Primary Antibody Diluent (MAGEA3/4, DNMT3A, and 3B) or Can Get Signal Solution A (5mC, DDX4, and DNMT3L) (TOYOBO). Primary antibodies used are 5mC (ab10805, Abcam, 1:50), MAGEA3/4 (MABC1150, Merck, 1:500),

DDX4 (AF2030, R&D, 1:500), DNMT3A (ab188470, Abcam, 1:500), DNMT3B (NB300-516-0.025 mg, Novus 1:200), DNMT3L (ab194094, Abcam, 1:150).

### Germ cell sorting
Monkey (human) testes were dissociated using 1 mg/mL collagenase type I (type IV) in DMEM at 37 °C with shaking for 30 min. After centrifugation (200 g, 5 min), the samples were digested with 0.25% trypsin-EDTA at 37 °C for 10 min. Reactions were halted by adding an equal volume of 10% FCS/DMEM. DNase I was added to a final concentration of 25 μg/mL, and the cells were incubated at room temperature for 1 min. After filtering through a 60-μm strainer, cells were centrifuged, and the supernatant was discarded. The washing process was repeated twice using 10% FCS/DMEM. Collagenase treatment was omitted for testes from younger animals (< 6 months old), with only trypsin treatment applied.

For fluorescence-activated cell sorting (FACS), monkey cells were stained with CD9-FITC (MCA469FT, BioRad, 1:50) and CD90-PE (555596, BD, 1:100) in 10% FCS/DMEM containing 25 μg/mL DNase1 for at least 30 min. Human cells were stained with SSEA4-Alexa488 (560308, BD, 1:100) in the same medium. DAPI was added to the FACS buffer (0.5% FCS/DMEM containing 25 μg/mL DNase1) to exclude dead cells. Sorting was performed using a SH800 (SONY). The proportion of germ cells in the FACS-purified population was confirmed via immunofluorescence with the MAGEA3/4 antibody (Fig. S8A, B). The germ cell proportions after sorting are shown in Supplementary Data 2.

### RNA and DNA purification from the same single cell
FACS-purified testicular cells were washed twice with 0.04% BSA/PBS and resuspended in 0.04% BSA/PBS for single-cell pickup. Single cells were isolated using a mouth pipette and placed into PCR tubes (MicroAmp 8-Tube Strip, Applied Biosystems). To each tube, 3.5 μL lysis buffer (1.14 U/μL RNAse Inhibitor, 0.54% Triton X-100) containing DynaBeads MyOne Carboxylic Acid (Thermo Fisher) was added. The beads bound to the nuclei, allowing separation of the cytoplasmic supernatant. For DNA extraction, 5 μL DNA isolation buffer (20 mM Tris-Cl, 2 mM EDTA, 20 mM KCl, 0.3% Triton X-100, 1 mg/mL proteinase K, 0.1 pg/μL λDNA, 2 ng/μL carrier RNA) was added to the beads, followed by incubation at 50 °C for 10 min. Proteinase K was inactivated at 75 °C for 30 min. scRNA-seq and scBS-seq libraries were then prepared from the cytoplasmic RNA and nuclear DNA fractions, respectively[18,30].

### scBS-seq and bulk BS-seq library preparation
scBS-seq libraries were prepared as per Zhou et al.[30]. with minor modifications. Unmethylated lambda DNA (Promega) was added prior to the bisulfite reaction. The MethylCode Bisulfite Conversion kit (Invitrogen) was used for bisulfite conversion of single-cell DNA. Four rounds of random priming were performed using a random primer with an A:T:G:C ratio of 4:4:1:1, minimizing GC-rich sequence amplification bias. After Exonuclease1 treatment to remove residual primers, second-strand synthesis and PCR amplification were conducted. Quality was assessed via a Bioanalyzer (Agilent), followed by paired-end 150-bp sequencing using Illumina NovaSeq.

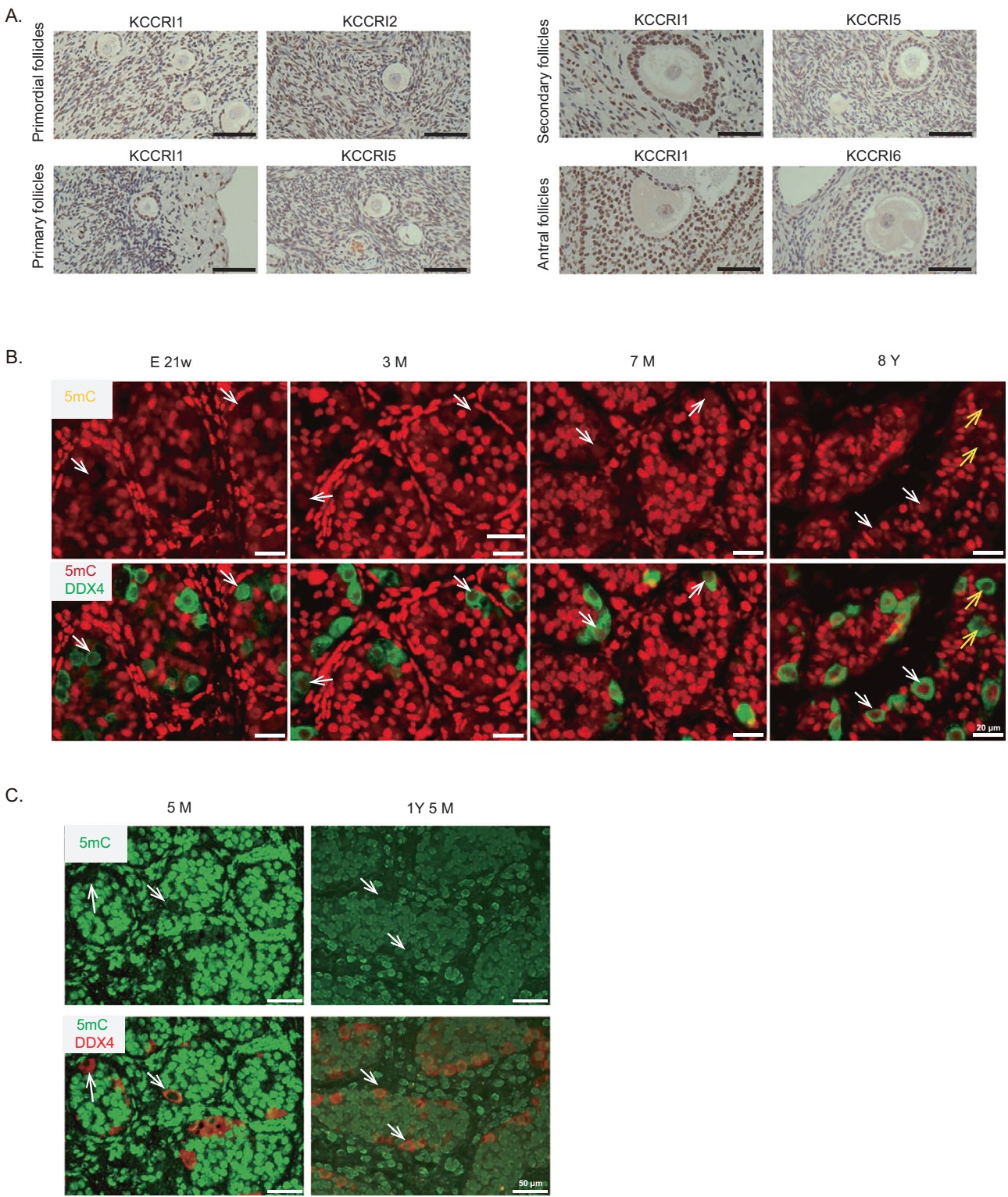

**Fig. 6 | Establishment of DNA methylation during human oocyte and testis development. A** Immunohistochemistry analyses of 5mC in ovaries from uterine cancer patients is shown. Sample IDs (KCCRI1, 2, 5, and 6) are indicated. Nuclei are counterstained with hematoxylin for clarity. Scale bars: 50 μm. **B** Immunofluorescence analyses of testes from aborted fetuses (21 weeks) and normal biopsy testes from children aged 3 months, 7 months, and 8 years are shown (see Supplementary Data 6 for sample information). Representative germ cells at each stage are marked with arrows. Yellow arrows in 8 Y pictures indicate possible false-negative cells. **C** Analysis of a normal portion of testes surgically removed for testicular teratoma with arrows indicating germ cells.

Bulk DNA was used for generating the sperm bisulfite-seq library (see Supplementary Methods for sperm isolation). The library was made with two modifications: (1) a single round of first-strand random priming with a biotinylated random primer and (2) purification of the first DNA strand using streptavidin beads, as per Miura et al.[31].

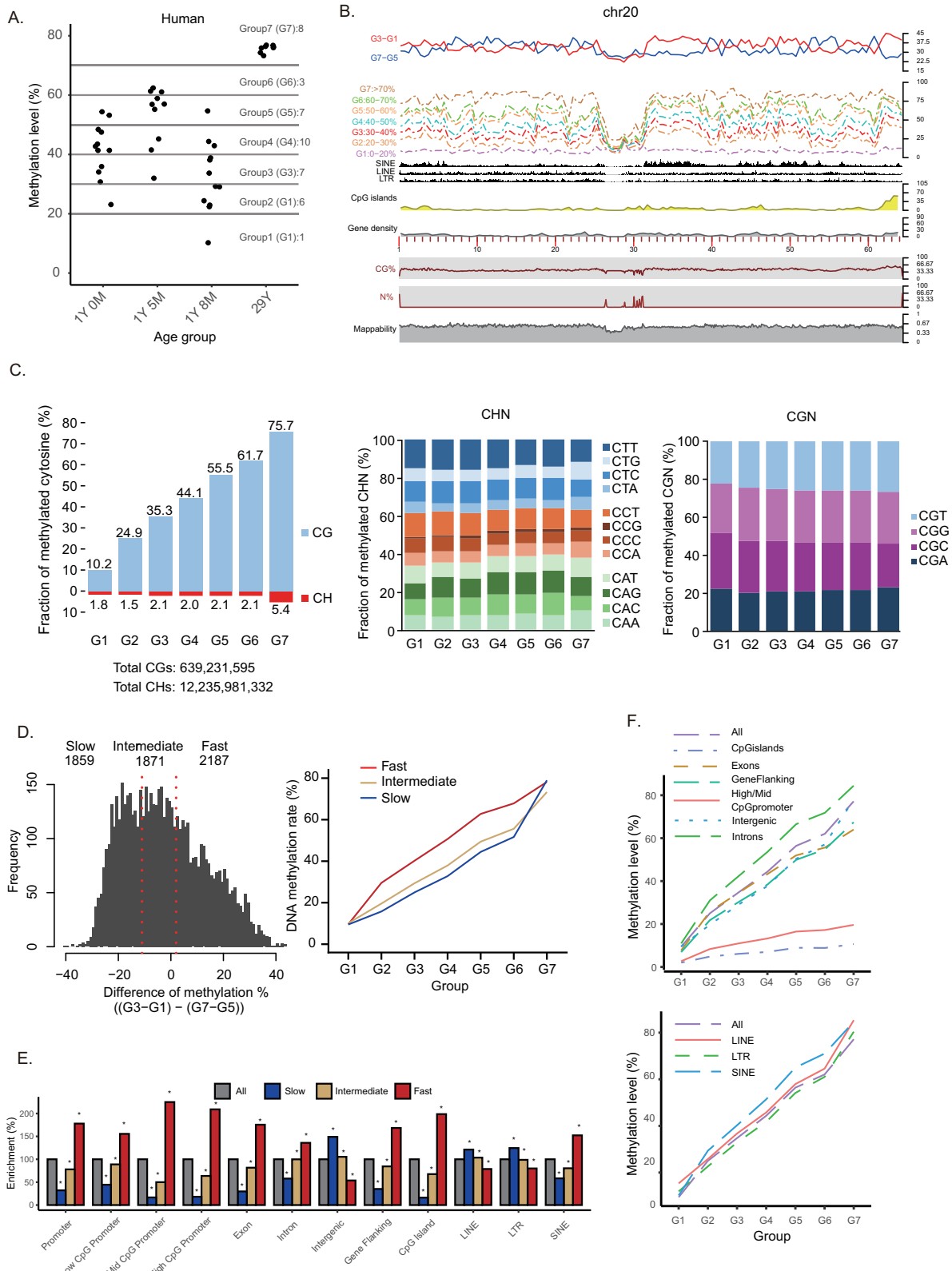

## scBS-seq data processing

Adapter and low-quality reads were trimmed using trim_galore (v.0.6.10) with parameters: --quality 20 --stringency 3 --length 50 --clip_R1 9 --clip_R2 9 --paired --trim1 --phred33. Clean reads were mapped to the marmoset genome (calJac4) using Bismark (v.0.24.2) in paired-end mode with parameters: --bowtie2 --fastq --non_directional. PCR duplicates were removed using the samtools (v.1.21) markup

command. Bismark methylation extractor, Bismark2bedGraph, and Coverage2cytosine analyses were performed sequentially for downstream processing. Among 103 marmoset cells, 30 cynomolgus monkey cells, and 46 human cells sequenced, we excluded 3, 0, and 4 cells, respectively. This exclusion was done based on two criteria: (1) coverage rate (< 1%) and (2) somatic cell contamination assessed via methylation levels at maternally methylated imprinted loci

**Fig. 7 | De novo DNA methylation in human male germ cells. A** DNA methylation levels during postnatal male germ cell development are shown, with each point representing the average methylation level of an individual cell measured by scBS-seq. Cells were grouped according to DNA methylation levels, with the number of cells in each group indicated to the right of the group labels. **B** Genomic annotations and the progression of DNA methylation establishment along chromosome 20 are depicted. **C** Fractions of methylated cytosines at CG (upper) and CH (lower) dinucleotides in prospermatogonia (G1–G6) and spermatogonial stem cells (G7) (left). The proportions of trinucleotide sequence motifs observed in methylated

CHN (center) and CGN (right) are shown. **D** Distinct de novo DNA methylation kinetics are illustrated. Genomic tiles (500 kbps) are grouped into three categories (slow, intermediate, fast) according to establishment patterns ((G3–G1) – (G7–G5), left panel). Average methylation trajectories for these classes are shown by group (right). **E** Enrichment of genomic annotations (e.g., CpG islands, TSSs) within the three classes is displayed. Asterisks show statistical significance ($P < 0.05$, Chi-squared test). **F** Average patterns of increasing DNA methylation levels across diverse genomic contexts. Source data are provided as a Source Data file for (**A**–**D**).

(MEST/PEG1, RB1, PLAGL1/ZAC1) (Supplementary Data 3, 5, and 7). The information of the imprinted loci is described in Supplementary Data 4.

Data from both DNA strands were merged to calculate CpG site methylation rates. The average methylation rate per cell was derived by averaging genome-wide CpG site methylation rates. Based on these averages, cells were classified into seven groups for marmosets (Group 1: 0%–10%, 11 cells; Group 2: 10%–20%, 16 cells; Group 3: 20%–30%, 8 cells; Group 4: 30%–40%, 9 cells; Group 5: 40%–50%, 22 cells; Group 6: 50%–60%, 19 cells; Group 7: 60%–80.1%, 15 cells), five groups for cynomolgus monkeys (Group 1: 2.5%–4.8%, 5 cells; Group 2: 28.4%–41.1%, 4 cells; Group 3: 52%–59.5%, 6 cells; Group 4: 66%–67.7%, 6 cells; Group 5: 68.2%–72.8%, 9 cells), and seven groups for humans (Group 1: 0%–20%, 1 cell; Group 2: 20%–30%, 6 cells; Group 3: 30%–40%, 7 cells; Group 4: 40%–50%, 10 cells; Group 5: 50%–60%, 7 cells; Group 6: 60%–70%, 3 cells; Group 7: 70%–80%, 8 cells).

### scRNA-seq library preparation

The scRNA-seq library preparation was conducted as previously described[18]. Before cDNA synthesis, ERCC RNA (Invitrogen, diluted 1:500,000) was added to the cytoplasmic RNA. cDNA was synthesized using cytoplasmic RNA, an oligo dT primer containing cell barcode sequences, template switch (TS) oligo DNA, and SuperScript II reverse transcriptase (Thermo Fisher). To prevent concatemerization of the TS oligo, nonnatural nucleotides were added to its 5′ end[32]. After purifying the cDNA with AMPure beads, qPCR was performed using primers detecting ERCC and the germ cell-specific gene DDX4 to evaluate the cDNA synthesis efficiency and confirm germ cell identity. cDNAs from 11 to 12 cells with different unique barcodes were pooled, and PCR was performed using a biotinylated primer targeting the 5′ end of the oligo dT sequence and an unmodified primer annealing to the 5′ end of the TS oligo. Fragmentation was carried out with the NEBNext Ultra II FS DNA Library Prep Kit for Illumina, and biotinylated DNA was purified using streptavidin beads. Adapter ligation and PCR amplification were performed according to the NEBNext Ultra II FS DNA Library Prep Kit protocol. Each library pool was labeled with a unique dual index (NEBNext Multiplex Oligos for Illumina, 96 Unique Dual Index). After quality assessment using Bioanalyzer (Agilent), paired-end 150-bp sequencing was performed on an Illumina NovaSeq platform.

### Statics & Reproducibility

The specific statistical test used, sample size (n), and P values are reported in the corresponding figure. Unless otherwise indicated, tests were two-sided, and a significance level of $α = 0.05$ was used. Reproducibility of the immunostaining results was described in the legends of the corresponding figures. The sample size for the scBS-seq data was first determined based on the predicted variation in methylation levels from immunofluorescence analyses. The linear increase pattern across developmental stages confirmed that our sample size is sufficient to understand overall developmental methylation dynamics.

### Reporting summary

Further information on research design is available in the Nature Portfolio Reporting Summary linked to this article.

## Data availability

scRNA/scBS/scATAC-seq data have been deposited in the DDBJ database under accession numbers: DRA016143 and DRA016233 and in the NBDC Human database under accession numbers: JGAS000887 (https://humandbs.dbcls.jp/en/hum0542-v1) and JGAS000888 (https://humandbs.dbcls.jp/en/hum0544-v1). Source data are provided with this paper.

## Code availability

Publicly available software and packages were mainly used to analyze scBS-seq data: for processing BAM files, Samtools version 1.21 (https://doi.org/10.1093/gigascience/giab008); for trimming adapter sequences, Trim Galore version 0.6.10 (https://zenodo.org/records/7598955), Cutadapt version 4.9 (https://doi.org/10.14806/ej.17.1.200), FastQC version 0.12.1; for mapping to the genome, Bismark version 0.24.2 (https://doi.org/10.1093/bioinformatics/btr167), Bowtie2 version 2.5.4 (https://doi.org/10.1109/SFCS.2000.892127); for annotation and analyzing data, bedtools version 2.31.0 (https://doi.org/10.1093/bioinformatics/btq033), Homer version 5.1 (https://doi.org/10.1016/j.molcel.2010.05.004); for displaying data on chromosomes, karyoploteR version 1.32.0 (https://doi.org/10.1093/bioinformatics/btx346), bismap version 1.2.1 (https://doi.org/10.1093/nar/gky677). Seurat version 5.1.0 (https://doi.org/10.1038/s41587-023-01767-y) and Cell Ranger were used to analyze scRNA-seq data. Cell Ranger and ArchR were used to analyze scATAC-seq data. The R and shell scripts used in the ATAC analyses can be obtained from GitHub (https://github.com/Hattyoriiiiiii/scatac-caljac-watanabe)[33]. Detailed information on analysis methods is provided in the respective sections of the Methods and Supplementary Methods.

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

## Acknowledgements

We thank Erika Sasaki and Akihiro Umezawa for support; Yoko Kuroki for mediation; Yoshiaki Kita, Erika Sasaki, and Tomomi Shimogori for the marmoset samples; Takayuki Mineshige, Keisuke Mukasa, Terumi Yurimoto, and Takashi Inoue for the help of surgery and veterinary care; Masatsugu Ema and Tomoyuki Tsukiyama for the cynomolgus monkey testes; Rui Wang and Fuchou Tang for information on generating scRNA-seq and scBS-seq libraries. Yoko Sato and Iwamoto Teruaki for samples. This work was supported by the following grants to T.W.: AMED PRIME (JP19gm6310010, JP20gm6310010, JP21gm6310010, and JP22gm6310010), KAKENHI (20H05764, 20H03177, 22K18356, 22H04923 (CoBiA) and 24KK0143), and JST (JPMJPR228B).

## Author contributions

K.K. and M.K.-I. prepared the scBS-seq and scRNA-seq libraries. Y.L., S.B.W., K.A. and L.B. analyzed the human samples. M.K. and T.S. collected the monkey samples. T.H. and S.M. performed the scATAC-seq analyses. S.T., Y.K., T.S., T.E. and R.N. performed the informatics analyses. T.Y., Y.H., Y.K., T.H., K.M. and H.K. provided the human testis samples. Y.M., S.S. and Y.D. provided the ovary samples. S.T. and T.W. wrote the paper. T.W. conceived the study.

## Competing interests

The authors declare no competing interests.

## Additional information

[1]Center for Regenerative Medicine, National Center for Child Health and Development (NCCHD), Tokyo, Japan. [2]Central Institute for Experimental Animals, Kawasaki, Japan. [3]Department of Urology, Center for Reproductive Sciences, University of California, San Francisco, San Francisco, USA. [4]Department of Histology and Cell Biology, Yokohama City University School of Medicine, Yokohama, Japan. [5]Tsukuba Primate Research Center, National Institutes of Biomedical Innovation, Health and Nutrition, Tsukuba, Japan. [6]Dynacom Inc, Chiba, Japan. [7]Division of Molecular Genetics and Epigenetics, Department of Biomolecular Sciences, Faculty of Medicine, Saga University, Saga, Japan. [8]Department of Urology, University of California, San Francisco, San Francisco, USA. [9]Division of Applied Biological Science, Faculty of Science and Technology, Tokyo University of Science, Noda, Japan. [10]Department of Growth and Reproduction, Copenhagen University Hospital-Rigshospitalet, Copenhagen, Denmark. [11]Department of Cellular and Molecular Medicine, Faculty of Health and Medical Sciences, University of Copenhagen, Copenhagen, Denmark. [12]Rhelixa Inc, Tokyo, Japan. [13]Molecular Pathology and Genetics Division, Kanagawa Cancer Center Research Institute, Yokohama, Japan. [14]Research Platform Office, Institute of Medical Science, The University of Tokyo, Tokyo, Japan. [15]Department of Medical Oncology and Cancer Center, and Center for Advanced Medicine against Cancer, Shiga University of Medical Science, Otsu, Japan. [16]Department of Pathology, NCCHD, Tokyo, Japan. [17]Division of Pediatric Urology, Department of Surgical Specialties, NCCHD, Tokyo, Japan. [18]Department of Pediatric Surgery, Niigata University Graduate School of Medical and Dental Sciences, Niigata, Japan. [19]Department of Pediatric Urology, Jichi Medical University, Children's Medical Center Tochigi, Shimotsuke, Japan. [20]Department of Urology, Toho University Faculty of Medicine, Tokyo, Japan. ✉e-mail: watanabe-tos@ncchd.go.jp

