## [Transparent Peer Review file · Nature Communications]

Establishment of DNA methylation during primate germ cell development

Corresponding Author: Dr Toshiaki Watanabe

Version 0:

Reviewer comments:

Reviewer #1

(Remarks to the Author)

The manuscript by Kojima et al. presents a detailed analysis of the timing and pattern of DNA methylation establishment during germ cell development across primate species. The authors utilize single-cell bisulfite sequencing (scBS-seq) alongside complementary scRNA-seq analyses to provide important insights into the dynamics of epigenetic reprogramming in primate germ cells, emphasizing species-specific differences compared to mouse models. The manuscript is concise and well-written; however, several points require attention before it can be recommended for publication. These points are outlined below.

While the authors mention in the abstract that the manuscript deals with three primate species: "we demonstrate that in three primate species - marmosets, macaques, and humans - de novo methylation occurs postnatally in prospermatogonia in males and growing oocytes in females", it is worth noting that the human data is very limited, devoid of sequencing controls, and in terms of sample size (of course completely understandable given the logistical and ethical difficulties in obtaining such human samples), not comparable to the datasets from marmosets and cynomolgus monkeys. While the human data might offer some preliminary insights and suggest conservation of the described mechanisms, further work is needed to unequivocally determine the dynamics of 5mC deposition in the postnatal germline.

While the study identifies differential 5mC kinetics between genic and intergenic regions, further mechanistic insight into why genic regions preferentially and more rapidly acquire methylation would enhance the impact. For example, could the authors utilize their RNA-seq data and classify genes according to expression levels to identify whether de novo 5mC in gene bodies correlates with increased transcriptional activity?

The work by Kubo et al. (BMC Genomics, 2015) reported very high levels of CH methylation (mCH) in rodent neonatal PSGs: "Approximately 10.8% of all cytosines were methylated in this cell type, compared with 3–4% in somatic tissues, and more than two-thirds of 5mCs (7.5% of all cytosines) were at CH sites (Fig. 2a)." Could the authors leverage their scBS-seq data (perhaps by pooling data to achieve greater depth) and analyze mCH? This would represent another important point the authors could easily address. The abundance and origin of the mCH signal, especially in the germline, has been much debated. The authors could use their data to demonstrate or dispute whether this is a conserved mechanism in mammals by quantifying mCH at various developmental stages and determining motif occurrence (e.g., CAC or CAG). The discovery that primate prospermatogonia undergo mitotic divisions during DNA methylation establishment, with DNMT1 expression implying methylation maintenance during mitosis, is particularly compelling. One could therefore hypothesize that mCH is diluted during these stages, as mCH mostly accumulates in non-dividing tissues. In any case, this would be an important analysis to perform.

Some quality control details for single-cell analyses (e.g., dropout rates, technical replicates, controls for bisulfite conversion efficiency) should be explained to strengthen the technical rigor of the reported findings.

Could the authors please clarify why certain germ cells showed negative staining for 5mC in older samples and discuss whether this might indicate technical issues or biological variability?

Reviewer #2

(Remarks to the Author)

In this manuscript, Kojima et al. investigate the timing of de novo DNA methylation during primate germ cell development, employing a combination of immunostaining, single-cell bisulfite sequencing (scBS-seq), and transcriptomic analyses. The study addresses an important question in the field of epigenetic reprogramming and offers valuable insights into methylation dynamics in non-human primates. The methodologies are largely appropriate, and the data presented are of high quality. However, the manuscript does not sufficiently address the mechanisms regulating the timing of methylation acquisition during germ cell development. Several critical issues need to be addressed to strengthen the manuscript and fully support the authors' conclusions.

Major Comments:

1. In Figures 1E and 1F, the authors demonstrate germ cell-specific acquisition of 5-methylcytosine (5mC). However, somatic cells appear to exhibit a progressive reduction in 5mC staining intensity with increasing age. This observation is not discussed or quantified. A systematic quantification of 5mC levels in somatic cells over developmental time points would help clarify whether global methylation loss in somatic cells accompanies germline reprogramming, which could have broader implications for understanding tissue-specific methylation dynamics.
2. In Figure 3B, cells are categorized into G1-G7 groups based on methylation levels. However, the manuscript does not adequately explain how these groupings correspond to developmental age or biological progression. As the classification is based on methylation states rather than chronological age, the authors should explicitly state this distinction. Including a schematic or trajectory plot mapping how cells from different developmental time points distribute across G1-G7 groups would enhance clarity and help contextualize the methylation progression relative to development.
3. The observation that genic regions preferentially gain methylation compared to intergenic regions is interesting. However, the role of chromatin accessibility (ATAC-seq peaks) in this process requires further explanation. Specifically, it is unclear whether ATAC signal intensity at earlier stages correlates with subsequent methylation acquisition. A more detailed description of how chromatin state influences methylation dynamics, and whether open chromatin regions are preferentially methylated during later stages (G5-G7), would substantially strengthen the mechanistic interpretation.
4. In Figure 3B, the authors report that DNMT3A is expressed across all groups, while DNMT3B and DNMT3L expression is more restricted to later stages. Interestingly, both DNMT3A and DNMT3B show biphasic expression patterns, with peaks at early (G2) and later (G5-G6) stages. This biphasic regulation is intriguing but is not discussed in the manuscript. A brief discussion exploring potential regulatory mechanisms underlying these expression patterns - such as the involvement of transcriptional regulators or cell cycle dynamics - would add depth to the interpretation.
5. While the transcriptomic data for DNMT3A, DNMT3B, and DNMT3L are informative, it is important to validate these findings at the protein level. Immunostaining or other protein-level analyses across developmental stages would provide critical validation of the temporal regulation of de novo methyltransferases.
6. In Figures 6 and 7, the authors assess methylation levels in oocytes and testicular cells obtained from uterine cancer patients and testicular biopsies, respectively. The use of cancer-derived samples may confound the interpretation of developmental methylation patterns due to potential alterations in the epigenetic landscape associated with malignancy. The authors should more thoroughly discuss the limitations of using cancer samples for developmental studies and, if possible, consider validating key findings in non-cancerous control samples.

Minor Comments:

Several figures would benefit from more detailed legends, explicitly stating the number of biological replicates and statistical analyses performed.

The manuscript would benefit from careful proofreading to correct minor grammatical errors and improve overall clarity.

Summary:

Overall, this manuscript addresses an important question and is based on impressive technical execution. However, significant revisions are necessary to strengthen the mechanistic insights and address potential confounding factors.

Reviewer #3

(Remarks to the Author)

Establishment of DNA methylation during germ cell development is well-characterized in mice, but our knowledge on this process in primates including humans is very limited. In this manuscript, Kojima et al. report their findings on the mode of DNA methylation acquisition in germ cells, mostly male germ cells, of marmosets, macaques, and humans although their molecular data are limited to marmosets and macaques. They find that prospermatogonia of primates undergo de novo methylation primarily during the first year after birth, unlike those of mice, where methylation is acquired during the fetal stage. Furthermore, they find that primate prospermatogonia undergo mitotic divisions, which indicates that establishment of methylation involves maintenance through DNA replication. These findings are novel and important for understanding the process of germ cell development in primates and have implications to fertility preservation for cancer patients, especially young male patients. Most of the mechanistic aspects remain to be elucidated, but I understand that such studies need time, especially for primates. Overall, the study provides important information. I should like to raise several points for improvement.

1. Lines 139-153: The authors discuss the pattern of DNA methylation acquisition across the genome by providing two hypothetical models (Fig. 3A). While I see the value of such models, they are not properly described in the text and, furthermore, not used in an effective way in deriving the conclusion. I request a more careful discussion with clear and correct wording.
2. Lines 283-289: The authors discuss possible mechanisms for the faster methylation acquisition in genic regions, but I was

surprised to find that they do not touch upon H3K36me. It is established that both DNMT3A and DNMT3B recognize H3K36me, which often marks genic and intergenic regions differently. Indeed, a study in mice showed that acquisition of methylation in male germ cells depends on H3K36me (Shirane et al. Nat. Genet. 2020). The authors should discuss this point.

3. Lines 320-339: The authors discuss possible mechanisms underlying the prolonged methylation acquisition in primates. While a difference in SAM concentration is an interesting possibility, this is totally speculative. I therefore suggest a more concise discussion on this point. On the other hand, I do not understand why they do not discuss their own finding, the lack of DAD-like regions in primates. The authors should discuss this point.

4. A problem of this manuscript is that the writing occasionally lacks smoothness, likely due to the authors' English writing ability. I strongly recommend English editing. Below are some points to be improved but there are many more.

- Line 43: To make this sentence meaningful, please add a brief phrase describing what is DNMT1.
- Line 80: Do the authors mean to describe methylation in the ovaries including somatic cells? Please correct it.
- Line 97: 5-methylcytosine and its abbreviation 5mC should be introduced here.
- Line 121: As scRNA-seq data do not appear in this section, it would be good to announce their appearance in later sections, by adding (see later).
- Line 148: A definition should be provided for the term "final level". Is it a methylation level at 12.5M or of G7?
- Lines 154-180: The words "genomic structures", "features", and "annotations" are intermingled in these paragraphs but are not properly used. Some sentences even have grammatical errors. Please tidy up.
- Line 195: "senescence G0 state" should be "quiescent G0 state"?

Version 1:

Reviewer comments:

Reviewer #1

(Remarks to the Author)

The authors have satisfactorily addressed all of my questions. The newly added analyses, including those on non-canonical DNA methylation, add substantial depth to the dataset and significantly enhance the value of this manuscript as a community resource.

Reviewer #2

(Remarks to the Author)

Based on the revisions described and the authors' point-by-point responses, the concerns raised have been satisfactorily addressed. The authors have:

Clarified and corrected the interpretation of 5mC immunostaining by repeating experiments with standardized exposure and improved sample quality.

Clearly explained the basis of the G1–G7 grouping and its relationship to methylation progression and developmental stages.

Strengthened the mechanistic interpretation of methylation acquisition by integrating chromatin accessibility and transcriptional data.

Appropriately addressed the apparent biphasic DNMT expression patterns and supported transcriptomic findings with new protein-level validation.

Transparently discussed the limitations associated with the use of cancer-derived samples.

Improved figure clarity, legends, and overall manuscript quality through careful editing.

Taken together, these revisions substantially strengthen the manuscript and fully support the authors' conclusions. I accept the manuscript in the present format.

Reviewer #3

(Remarks to the Author)

The authors have appropriately addressed almost all of the points I raised. Furthermore, the addition of scBS-seq data for human prospermatogonia and spermatogonial stem cells is impressive and further strengthens the manuscript.

One minor point: the clause added in the Abstract to explain the maintenance methyltransferase DNMT1 ("..., which adds a methyl group to the newly synthesized DNA strand using the information regarding the methylation status of the template strand, ...") seems overly detailed and could be deleted.

Point-by-point response to the reviewer's comments.

The reviewer's comments are presented in blue font, and our replies are described in red.

REVIEWER COMMENTS

Reviewer #1 (Remarks to the Author):

The manuscript by Kojima et al. presents a detailed analysis of the timing and pattern of DNA methylation establishment during germ cell development across primate species. The authors utilize single-cell bisulfite sequencing (scBS-seq) alongside complementary scRNA-seq analyses to provide important insights into the dynamics of epigenetic reprogramming in primate germ cells, emphasizing species-specific differences compared to mouse models. The manuscript is concise and well-written; however, several points require attention before it can be recommended for publication. These points are outlined below.

① While the authors mention in the abstract that the manuscript deals with three primate species: "we demonstrate that in three primate species - marmosets, macaques, and humans - de novo methylation occurs postnatally in prospermatogonia in males and growing oocytes in females", it is worth noting that the human data is very limited, devoid of sequencing controls, and in terms of sample size (of course completely understandable given the logistical and ethical difficulties in obtaining such human samples), not comparable to the datasets from marmosets and cynomolgus monkeys. While the human data might offer some preliminary insights and suggest conservation of the described mechanisms, further work is needed to unequivocally determine the dynamics of 5mC deposition in the postnatal germline.

Response: In the revised version, we incorporated scBS-seq data for human prospermatogonia and spermatogonial stem cells (lines 366-389, Fig 7). This analysis confirms that human germ cells exhibit a similar annotation-dependent DNA methylation establishment kinetics to those of monkey germ cells. However, the establishment in humans appears to begin later and take longer. Some of the samples examined here were derived from testes with teratoma; their developmental progression may be affected. Future studies using normal tissues are required to confirm the results. We clearly described this in the revised version. In addition, we added one testicular sample for immunostaining analysis of 5-mC (3-month-old testicular sample from a testicular tumor (hemangioma) patient, Figure 6D). This analysis confirmed that de novo methylation is initiated at 3-month-old testes. Adding human sequencing data and new immunostaining data increased the significance of our paper. We thank the reviewer for the comment.

② While the study identifies differential 5mC kinetics between genic and intergenic regions, further mechanistic insight into why genic regions preferentially and more rapidly acquire methylation would enhance the impact. For example, could the authors utilize their RNA-seq data and classify genes according to expression levels to identify whether de novo 5mC in gene bodies correlates with increased transcriptional activity?

Response: We thank the reviewer for this valuable suggestion. We analyzed our RNA-seq data from marmoset prospermatogonia and classified genes into three categories based on expression levels—high, mid, and low—to examine their methylation status. This analysis revealed that DNA methylation levels in genic regions are positively correlated with transcription levels: highly expressed genes showed higher DNA methylation levels within gene bodies than weakly expressed genes. This finding is consistent with the previous studies showing that de novo DNA methylation in oocytes depends on transcriptional activity. We have incorporated these results into the revised manuscript (lines 214–228, Fig. 3H and I).

③ The work by Kubo et al. (BMC Genomics, 2015) reported very high levels of CH methylation (mCH) in rodent neonatal PSGs: "Approximately 10.8% of all cytosines were methylated in this cell type, compared with

3–4% in somatic tissues, and more than two-thirds of 5mCs (7.5% of all cytosines) were at CH sites (Fig. 2a)." Could the authors leverage their scBS-seq data (perhaps by pooling data to achieve greater depth) and analyze mCH? This would represent another important point the authors could easily address. The abundance and origin of the mCH signal, especially in the germline, has been much debated. The authors could use their data to demonstrate or dispute whether this is a conserved mechanism in mammals by quantifying mCH at various developmental stages and determining motif occurrence (e.g., CAC or CAG). The discovery that primate prospermatogonia undergo mitotic divisions during DNA methylation establishment, with DNMT1 expression implying methylation maintenance during mitosis, is particularly compelling. One could therefore hypothesize that mCH is diluted during these stages, as mCH mostly accumulates in non-dividing tissues. In any case, this would be an important analysis to perform.

Response: We thank the reviewer for raising this important point. As suggested, we analyzed CH methylation levels in marmoset, cynomolgus monkey, and human prospermatogonia. Our analysis demonstrated that CH methylation remained consistently low in primate prospermatogonia (marmoset: 0.7%–1.1%; cynomolgus: 1.0%–1.8%; humans: 1.5–2.1%). This contrasts sharply with the situation in mouse prospermatogonia (7.5%) and highlights species-specific features of CH methylation during this stage of germ cell development.

Our detailed analysis further revealed that methylated CH motifs are enriched for CA dinucleotides, showing an increasing trend during development. Moreover, the CAC and CAG trinucleotides were the most prevalent motifs among methylated CA sequences. Since the preferred target sequences for DNMT3A and DNMT3B are CAC and CAG, respectively, these findings suggest that CH methylation in primate prospermatogonia depends on DNMT3A/DNMT3B activity. Consistently, we confirmed an increasing trend in DNMT3A and DNMT3B mRNA levels, correlating with the rise in mCH levels.

As proposed by the reviewer, the substantial differences between mouse and primates may be explained by the fact that primate prospermatogonia remain mitotically active at this stage. We have incorporated these results into the revised manuscript (lines 267–285, 316–321, and 387–394, Fig. 4E, F, 5C, D, 7C).

④ Some quality control details for single-cell analyses (e.g., dropout rates, technical replicates, controls for bisulfite conversion efficiency) should be explained to strengthen the technical rigor of the reported findings.

Response: As suggested, we have added the information on dropout rates, number of replicates at each stage, conversion rates, and genome coverage rates necessary for evaluating scBS-seq data quality (lines 136–142). We believe that including these details further demonstrates the accuracy and reliability of our dataset.

⑤ Could the authors please clarify why certain germ cells showed negative staining for 5mC in older samples and discuss whether this might indicate technical issues or biological variability?

Response: In the revised version, we added new data showing that long-term storage results in similar negative staining. We compared the staining pattern before and after storage for several months. Negative staining was observed only after staining. These data and the discussions of the data have been incorporated into the manuscript and the supplementary figure (lines 359–369, Figure S6).

Reviewer #2 (Remarks to the Author):

In this manuscript, Kojima et al. investigate the timing of de novo DNA methylation during primate germ cell development, employing a combination of immunostaining, single-cell bisulfite sequencing (scBS-seq), and transcriptomic analyses. The study addresses an important question in the field of epigenetic reprogramming and offers valuable insights into methylation dynamics in non-human primates. The methodologies are largely appropriate, and the data presented are of high quality. However, the manuscript does not sufficiently address the mechanisms regulating the timing of methylation acquisition during germ cell development. Several critical issues need to be addressed to strengthen the manuscript and fully support the authors' conclusions.

Major Comments:

1. In Figures 1E and 1F, the authors demonstrate germ cell-specific acquisition of 5-methylcytosine (5mC). However, somatic cells appear to exhibit a progressive reduction in 5mC staining intensity with increasing age. This observation is not discussed or quantified. A systematic quantification of 5mC levels in somatic cells over developmental time points would help clarify whether global methylation loss in somatic cells accompanies germline reprogramming, which could have broader implications for understanding tissue-specific

methylation dynamics.

Response: We thank the reviewer for highlighting this issue. The images shown in Figures 1E and 1F were indeed prone to misinterpretation. To improve visibility of weak 5mC signals in germ cells, we had increased the exposure time of early-age samples. We believe that these differences in exposure time, combined with variations in tissue quality, may have led to a misunderstanding of the methylation status of somatic cells. To resolve this, we repeated the immunostaining and captured images using identical exposure times. Furthermore, we replaced the lower-quality samples with higher-quality ones (marmoset 33-day-old testis → 21-day-old testis; cynomolgus monkey 1-year- and 11-month-old testis → 1-year- and 9-month-old testis; Figure 1E, 1F, and Table S1). We believe the new images more accurately represent actual DNA methylation levels.

2. In Figure 3B, cells are categorized into G1-G7 groups based on methylation levels. However, the manuscript does not adequately explain how these groupings correspond to developmental age or biological progression. As the classification is based on methylation states rather than chronological age, the authors should explicitly state this distinction. Including a schematic or trajectory plot mapping how cells from different developmental time points distribute across G1-G7 groups would enhance clarity and help contextualize the methylation progression relative to development.

Response: We thank the reviewer for this comment. As suggested, we have revised the manuscript (lines 160–167) and the corresponding figure (Figure 2, 5A, and 7A) to clarify the relationships among grouping, methylation levels, and developmental stages.

3. The observation that genic regions preferentially gain methylation compared to intergenic regions is interesting. However, the role of chromatin accessibility (ATAC-seq peaks) in this process requires further explanation. Specifically, it is unclear whether ATAC signal intensity at earlier stages correlates with subsequent methylation acquisition. A more detailed description of how chromatin state influences methylation dynamics, and whether open chromatin regions are preferentially methylated during later stages (G5-G7), would substantially strengthen the mechanistic interpretation.

Response: We appreciate the reviewer's suggestion. We analyzed ATAC-seq data from marmoset prospermatogonia to assess the relationship between chromatin accessibility and DNA methylation acquisition at genic regions. This analysis revealed that "fast" genes display greater promoter accessibility than "intermediate" and "slow" genes (Figure 3I, lines 222–226). This finding is consistent with our additional results showing that methylation acquisition at gene bodies positively correlates with transcription levels (Figure 3H, lines 219–222). These new results have been incorporated into the revised manuscript.

4. In Figure 3B, the authors report that DNMT3A is expressed across all groups, while DNMT3B and DNMT3L expression is more restricted to later stages. Interestingly, both DNMT3A and DNMT3B show biphasic expression patterns, with peaks at early (G2) and later (G5-G6) stages. This biphasic regulation is intriguing but is not discussed in the manuscript. A brief discussion exploring potential regulatory mechanisms underlying these expression patterns - such as the involvement of transcriptional regulators or cell cycle dynamics - would add depth to the interpretation.

Response: We consider the biphasic pattern observed to be largely attributable to technical variation resulting from the limited number of cells and the small number of UMIs per cell. We have added this explanation to the revised version (lines 235–237).

5. While the transcriptomic data for DNMT3A, DNMT3B, and DNMT3L are informative, it is important to validate these findings at the protein level. Immunostaining or other protein-level analyses across developmental stages would provide critical validation of the temporal regulation of de novo methyltransferases.

Response: We appreciate this comment. We performed immunofluorescence analyses as suggested. These analyses support the overall expression patterns revealed by scRNA-seq and also provide two new insights. First, they showed substantial variation in expression levels among individual cells. Second, they suggested that DNMT3A and DNMT3B may be regulated post-transcriptionally through nuclear import and export. These data have been added to the revised manuscript (lines 239–249, Figure 4C).

6. In Figures 6 and 7, the authors assess methylation levels in oocytes and testicular cells obtained from uterine cancer patients and testicular biopsies, respectively. The use of cancer-derived samples may confound the interpretation of developmental methylation patterns due to potential alterations in the epigenetic landscape associated with malignancy. The authors should more thoroughly discuss the limitations of using cancer samples for developmental studies and, if possible, consider validating key findings in non-cancerous control samples.

Response: We agree with this comment. We used several cancer patient-derived tissues for methylation analyses. The potential impact of cancer on testis/ovary development depends on the cancer type. We consider that testicular teratoma may affect the developmental progression of testicular tissues because of their close proximity. By contrast, other cancers originated outside the testes/ovaries, and therefore the associated risk should be minimal. We have added sentences in the revised version explaining these limitations for each cancer type (lines 335–338, 348–353, 380–383, and 407–409).

Minor Comments:

Several figures would benefit from more detailed legends, explicitly stating the number of biological replicates and statistical analyses performed.

The manuscript would benefit from careful proofreading to correct minor grammatical errors and improve overall clarity.

Response: In accordance with the reviewer's comments, we carefully reviewed the manuscript, figures, and legends. Detailed information—including the number of replicates and the statistical analyses performed—was added to the figures and their legends. In addition, the manuscript was edited to correct grammatical errors and enhance overall readability.

Summary:

Overall, this manuscript addresses an important question and is based on impressive technical execution. However, significant revisions are necessary to strengthen the mechanistic insights and address potential confounding factors.

Response: We thank the reviewer for recognizing the importance of our study and for the positive evaluation of our methodology.

Reviewer #3 (Remarks to the Author):

Establishment of DNA methylation during germ cell development is well-characterized in mice, but our knowledge on this process in primates including humans is very limited. In this manuscript, Kojima et al. report their findings on the mode of DNA methylation acquisition in germ cells, mostly male germ cells, of marmosets, macaques, and humans although their molecular data are limited to marmosets and macaques. They find that prospermatogonia of primates undergo de novo methylation primarily during the first year after birth, unlike those of mice, where methylation is acquired during the fetal stage. Furthermore, they find that primate prospermatogonia undergo mitotic divisions, which indicates that establishment of methylation involves maintenance through DNA replication. These findings are novel and important for understanding the process of germ cell development in primates and have implications to fertility preservation for cancer patients, especially young male patients. Most of the mechanistic aspects remain to be elucidated, but I understand that such studies need time, especially for primates. Overall, the study provides important information. I should like to raise several points for improvement.

1. Lines 139–153: The authors discuss the pattern of DNA methylation acquisition across the genome by providing two hypothetical models (Fig. 3A). While I see the value of such models, they are not properly described in the text and, furthermore, not used in an effective way in deriving the conclusion. I request a more careful discussion with clear and correct wording.

Response: We appreciate the reviewer for highlighting this issue. As suggested, we extensively revised this section. We believe that, in the revised version, the hypotheses facilitate a clearer understanding of the results and contribute to a stronger conclusion (lines 168–182).

2. Lines 283–289: The authors discuss possible mechanisms for the faster methylation acquisition in genic

regions, but I was surprised to find that they do not touch upon H3K36me. It is established that both DNMT3A and DNMT3B recognize H3K36me, which often marks genic and intergenic regions differently. Indeed, a study in mice showed that acquisition of methylation in male germ cells depends on H3K36me (Shirane et al. Nat. Genet. 2020). The authors should discuss this point.

Response: As suggested by the reviewer, we now discuss the potential involvement of H3K36me_{2/3} marks in differential methylation between genic and intergenic regions (lines 423–430, Figure 4A). We thank the reviewer for this valuable input.

3. Lines 320-339: The authors discuss possible mechanisms underlying the prolonged methylation acquisition in primates. While a difference in SAM concentration is an interesting possibility, this is totally speculative. I therefore suggest a more concise discussion on this point. On the other hand, I do not understand why they do not discuss their own finding, the lack of DAD-like regions in primates. The authors should discuss this point.

Response: We appreciate the reviewer's thoughtful suggestion. Accordingly, we have shortened the discussion on SAM concentration and, instead, added discussion on DAD-like regions in the revised version (lines 463–476).

4. A problem of this manuscript is that the writing occasionally lacks smoothness, likely due to the authors' English writing ability. I strongly recommend English editing. Below are some points to be improved but there are many more.

- Line 43: To make this sentence meaningful, please add a brief phrase describing what is DNMT1.

- Line 80: Do the authors mean to describe methylation in the ovaries including somatic cells? Please correct it.

- Line 97: 5-methylcytosine and its abbreviation 5mC should be introduced here.

- Line 121: As scRNA-seq data do not appear in this section, it would be good to announce their appearance in later sections, by adding (see later).

- Line 148: A definition should be provided for the term "final level". Is it a methylation level at 12.5M or of G7?

- Lines 154-180: The words "genomic structures", "features", and "annotations" are intermingled in these paragraphs but are not properly used. Some sentences even have grammatical errors. Please tidy up.

- Line 195: "senescence G0 state" should be "quiescent G0 state"?

Response: We sincerely thank the reviewer for these comments. The manuscript has been carefully edited to improve readability and accuracy.